# Influence of Walking as Physiological Training to Improve Respiratory Parameters in the Elderly Population

**DOI:** 10.3390/ijerph19137995

**Published:** 2022-06-29

**Authors:** Klára Novotová, Dagmar Pavlů, Dominika Dvořáčková, Anna Arnal-Gómez, Gemma Victoria Espí-López

**Affiliations:** 1Faculty of Physical Education and Sport, Charles University, 162 52 Prague, Czech Republic; novotova.klara@ftvs.cuni.cz (K.N.); dvorackova.dominika@ftvs.cuni.cz (D.D.); 2Department of Physiotherapy, Faculty of Physiotherapy, University of Valencia, 46010 Valencia, Spain; anna.arnal@uv.es (A.A.-G.); gemma.espi@uv.es (G.V.E.-L.); 3Physiotherapy in Motion, Multispecialty Research Group (PTinMOTION), 46010 Valencia, Spain; 4Exercise Intervention for Health (EXINH), University of Valencia, 46010 Valencia, Spain

**Keywords:** walking, aerobic training, elderly, older adults, spirometry, FVC, FEV_1_, lung function

## Abstract

Walking as physiological training is reported to be an effective activity in order to beneficially influence and slow the onset of aging in healthy elderly people. However, insufficient evidence exists on how walking influences lung function in seniors. In our study, we aim to evaluate the effect of different types of walking on lung function in healthy seniors. The PubMed, Web of Science, Scopus and EBSCO Essentials databases were searched, while the methodological quality was assessed by the RoB2 tool. A total of seven studies (RCTs) published between 2002–2022 that met the eligibility criteria were analysed in this review. All participants were older adults without any specific associated disease, aged 60 and above. The interventions included structured physical activity; a high/moderate exercise program; long-term regular walking; walking as a part of functional movement training; walking sideways, backward and forward as a part of aerobic training; fast walking; Stepper walking; walking on a treadmill combined with incentive spirometry; and Nordic walking. Overall, most of the mentioned types of walking led to improved lung function in healthy elderly subjects. However, the prescribed Stepper walking program did not improve lung function in healthy seniors.

## 1. Introduction

The worldwide population is tending to rapidly age. Numerous statistics show that the number of people aged 60 and above will rise in the future. The World Health Organisation stated that, in 2019, this number reached one billion, and we expect this number to rise to 2.1 billion in 2050 [1]. The population should respond to these predictions, among other things, by developing exercise programs that could slow down the onset of signs of ageing [2]. Age-related changes affect every system of the body, including the respiratory system [3]. Lung fibrosis is present and negatively impacts breathing [4]. The inspiratory muscles weaken, thereby the effectiveness of coughing in airway clearance is reduced. The weakening of the inspiratory muscles often occurs because of sarcopenia, a condition in which there is a gradual loss of skeletal muscle mass in the body [5,6]. As sarcopenia progresses, physical fitness decreases in elderly people, which may later be reflected in limited chest and overall mobility [7,8].

With increasing age, breathing is influenced by several factors that usually affect it negatively. Kyphotic posture and forward head posture negatively impact respiration in older adults [9,10]. Inappropriate head posture adversely affects the respiratory mechanics, making it less effective [11]. When a forward head posture is present, diaphragmatic breathing is limited; on the contrary, the auxiliary inspiratory muscles are over-activated, which leads to their overuse [9]. This persistent hyperactivity of the auxiliary inspiratory muscles is disadvantageous for the organism, not only in terms of respiratory mechanics, but also in terms of energy in general [12].

All the age-related changes in the respiratory system can be adequately evaluated by spirometry [13]. In terms of spirometry parameters, there is a decrease in the static and dynamic ventilation parameters of the lungs and airways in seniors. This is mainly a consequence of the changes to the respiratory system mentioned above [3,14,15].

Old age as such does not directly cause any particular disease [16]. However, some studies state that we can observe limited physical activity in many seniors [17,18]. This limitation was greater during the COVID pandemic [19]. There is evidence that various physical therapy interventions can reverse the negative impact of old age on individual’s overall health [20,21,22,23]. Among other physical activities, leisure walking is considered the most cost-effective exercise as it does not require any equipment or skill [24].

Walking as physiological training is reported to be one of the appropriate activities to beneficially influence and slow the onset of aging in healthy elderly people [25,26,27,28,29]. These benefits include, for example, a reduction in resting heart rate [26], improved postural stability [25], improved subjective perception of fatigue [26], increased self-selected walking speed [29], reduced anxiety [30] and improved overall mental health [31,32]. It is therefore appropriate for seniors to walk regularly, thus contributing to their health in physical, mental and social terms [33].

However, insufficient evidence exists on how walking influences respiratory parameters in seniors. In our study, we aim to evaluate the effect of different types of walking, as physiological physical activity, on respiratory outcomes in healthy seniors.

## 2. Materials and Methods

The current review was written according to the Preferred Reporting Items for Systematic Reviews and Meta-Analysis checklist [34,35]. The PRISMA statement for this study can be found in Table 1. To provide credible evidence, we followed the guidelines of the Cochrane Handbook for Systematic Reviews of Interventions version 6.3. [36]. The authors tried to collect all the compliant studies and present a search strategy to obtain reproducible search results. Studies that met the eligibility criteria of this review were included in the study, and their results were subsequently synthesized. All the studies analysed in this review had to meet the inclusion criteria. The quality of each study, included in the review, was assessed using the Cochrane risk of bias tool version 2 (RoB2) [36].

### 2.1. PICOS and Eligibility Criteria

The research question was set by specifying the Population, Intervention, Comparison, Measured Outcome, and Study Design (PICOS) framework. To classify studies relevant to this review, the eligibility criteria based on the PICOS framework and study questions were determined (Table 2).

### 2.2. Search Strategy

The keywords used to search the relevant studies that could be analysed in this review were established based on various studies dealing with walking intervention in elderly healthy people [37,38,39,40,41]. This review was limited to analysing randomized controlled trials only. Keywords (Table 3) to identify intervention, participants, comparators, measured outcomes and study design were entered in various combinations into multiple databases’ search box. These databases included PubMed, Web of Science, Scopus and EBSCO Essentials. All studies were published in English in the period from February 2005 to January 2019. Initial screening was undertaken by one researcher (K.N.) and then checked independently by another researcher (D.D.). Disagreements about inclusion were discussed until agreement was reached. One researcher (K.N.) extracted the data from the selected studies. Another researcher (D.P.) verified the extracted data and made corrections where necessary. Disagreements were resolved by reference to the third and fourth researcher (A.A.G. and G.E.L.). All researchers contributed to the synthesis of the data. The level of studies was limited to randomized controlled trials (RCTs). The authors sought to identify all relevant studies that involved walking as a means of influencing respiratory parameters in the elderly population.

### 2.3. Study Selection

From the database search, we found a total of 2959 records. Of these, 248 were found on Web of Science, 390 on PubMed, 1652 on Scopus, and 669 on EBSCO Essentials. After tightening the search criteria in terms of language (English), type of publication (article) and publication years (2002–2022), the titles and abstract of 2405 studies were analysed. Two reviewers independently screened (K.N. and D.D.) the full text for 124 documents based on the eligibility criteria. The final number of studies included in the review was 7 (Figure 1).

### 2.4. Inclusion Criteria

Only studies that met the following criteria were analysed in this review:-Studies working with seniors without a specific comorbid disease;-Studies analysing different types of walking (also as a part of a more comprehensive physical intervention) in seniors;-Studies working with seniors aged 60 and above;-Studies designed as a randomized controlled trial (RCT) or randomized cross-over trial;-Studies working with probands that do not suffer from any other disease that could affect the results of the study;-Studies working with probands that were able to provide written consent to participate in the study;-Studies published in the English language;-Studies published between 2002 and 2022.

### 2.5. Exclusion Criteria

Studies that did not meet any of the conditions listed in the inclusion criteria were considered non-compliant and were excluded from this review. Most of the studies that were discarded during the study of the literature sources dealt with the issue of walking in seniors with a specific associated disease.

### 2.6. Data Extraction

Data were extracted from 7 selected documents. The extracted data included study information (year of publication, author), samples (sample size, gender, characteristics of participants), intervention design (activity duration, activity frequency), study design, outcome measures and results.

### 2.7. Assessing the Risk of Bias

The quality of studies analysed in this review was investigated by assessing the risk of bias in each study. The risk of bias was assessed using the Cochrane risk of bias tool in the framework of its latest version. The RoB2 tool was used because it provides an appropriate way to assess the risk of bias in randomized controlled trials. The RoB2 tool also enables identifying various forms of bias that may reveal bias during randomization, dropout, measurement and selecting results, and it can also detect missing data [38].

## 3. Results

### 3.1. Study Characteristics

A total of 1857 participants were reported in the seven included studies, with a sample size ranging from 30 to 1635. The largest sample size (1635) was observed in the multicentre, single-blind, parallel randomized trial. In the remaining six studies, the sample size ranged from 30 to 45. It was not possible to assess the precise summary of women and men enrolled in the studies. One of the studies did not mention the precise number of women and men participating in it. The age of participants ranged approximately from 60 to 88 years. The main characteristics of the included studies are summarized in Table 4.

The intervention activities conducted in the studies included structured physical activity (walking + strength, flexibility and balance training), a high/moderate exercise program (fast walking in the aerobic part + stretching, flexibility exercises), long-term regular walking, walking as a part of functional movement training, walking sideways/backward and forward as a part of aerobic training alone or combined with nutritional intervention, fast walking, Stepper walking, walking on treadmill combined with incentive spirometry and Nordic walking. The duration of the intervention in the studies ranged from 8 weeks to 42 months.

### 3.2. The Outcome Measures

The outcome measures in the studies included:-Lung function: forced vital capacity (FVC), forced expiratory volume in 1 s (FEV_1_), vital capacity (VC), maximal voluntary ventilation (MVV), maximum inspiratory mouth pressure (PImax. or MIP), maximum expiratory mouth pressure (PEmax.);-Cardiac function: resting heart rate, systolic blood pressure, diastolic blood pressure;-General information: weight, height, body mass index (BMI), waist/hip ratio, body fat percentage;-Muscle strength: handgrip strength, handgrip/body weight ratio.-Endurance: aerobic endurance, inspiratory muscle endurance, calf endurance.-Physical activity: physical activity level, fatigue intensity in performing ADL and IADL;-Other outcome measures: six-minute walk test (6MWT), exacerbation of obstructive airway disease (EOAD), reaction time, flexibility, Sit and Reach test, Get Up and Go test, the functional movement screen (FMS), quality of life (SF-36), Mini mental state exam (MMSE), Barthel index, Instrumental Activities of Daily Living (IADL), Geriatric Depression Scale (GDS).

#### 3.2.1. Lung Function

All seven studies analysed in this review included spirometry as an evaluation tool to assess the lung function. These studies dealt with forced vital capacity (FVC), vital capacity (VC), 1 s forced expiratory volume (FEV_1_) and maximal voluntary ventilation (MVV). MIP and MEP were assessed by using the mechanical pressure gauge. None of the above respiratory parameters were present in all studies. In terms of spirometry, FEV_1_ was the most common measured outcome occurring in a total of 5 studies out of 7, followed by FVC, which occurred in a total of 4 studies out of 7. Other parameters occurred on average in two studies (VC, MVV, MIP/PImax.), and MEP occurred in one study.

There was a statistically significant improvement of FEV_1_ in the intervention group following the intervention in three studies [39,42,43]. These studies included a high-/moderate-intensity exercise program [42], aerobic training with rhythmic functional movement [43] and aerobic exercise training combined with incentive spirometry [38]. The two studies [38,42] showed no statistically significant improvement of FEV_1_ in the control group. However, FEV_1_ improved in the control group in the study of Shim et al. This improvement was observed within the group. No statistically significant improvement of FEV_1_ was observed in the intervention group nor the control group post-intervention in two studies [38,44]. These studies included structured physical activity [44] and a prescribed Stepper walking program [38].

FVC was significantly improved in the intervention group in a total of three studies including a high-/moderate-intensity exercise program [42], long-term regular walking [41] and aerobic training combined with rhythmic functional movement [43]. These studies did not show any improvement of FVC in the control group. There was no significant improvement of FVC in the study including the prescribed Stepper walking program [39].

MVV significantly improved in the intervention group, but not in the control group in the study including aerobic exercise training combined with incentive spirometry [38] MVV also significantly improved in the study analysing aerobic training with rhythmic functional movement [43]; nonetheless, the improvement occurred in the intervention group, as well as in the control group. This improvement was observed within each group, but also between the two groups.

MIP was measured in two studies including structured physical activity [44] and aerobic training combined with incentive spirometry [38]. MIP was significantly improved in the intervention group and not in the control group in the study of El-Kader et al. [38]. In the study by Fragoso et al. [44], there was no significant improvement of MIP whatsoever. MEP occurred in the study including aerobic training combined with incentive spirometry [38], where MEP was significantly improved in the intervention group, but not in the control group.

VC was significantly improved in two studies examining Nordic walking and aerobic exercise training combined with incentive spirometry [38,45].

To conclude, FEV_1_ significantly improved in participants of the intervention group in 3 of 5 studies including a high-/moderate-intensity exercise program [42], aerobic training with rhythmic functional movement [43] and aerobic exercise training combined with incentive spirometry [38]. FVC improved in participants of the intervention group in 3 of 4 studies including a high-/moderate-intensity exercise program [42], long-term regular walking [41] and aerobic training combined with rhythmic functional movement [43]. This review mainly focuses on the respiratory outcomes (FVC, FEV1_1_) and their change over time in each study.

#### 3.2.2. Cardiac Function

Cardiac function was assessed based on resting heart rate and blood pressure (both systolic and diastolic), and all these outcomes occurred in the study analysing a high-/moderate-intensity exercise program [42]. There was no significant improvement in the time of resting heart rate in any group. The information on the blood pressure change is not included in the results of the study.

#### 3.2.3. General Information

The general information includes outcomes such as weight, height, body mass index (BMI), waist/hip ratio and body fat percentage. Weight and height were assessed in the total of three studies [41,44,45]. As the information on height and its change in time were not specified in any study, for the purpose of this review, we can assume that the height of the participants did not significantly change in any of the three studies. Weight significantly deceased in the intervention group, but not in the control group of the Nordic walking study [45]. Weight did not change in the participants of the study including a high-/moderate-intensity exercise program [42]. Saygin et al. [41] did not mention whether the change in height and weight was statistically significant.

The body mass index (BMI) was assessed in three studies. BMI did significantly decrease in the intervention group and not in the control group in two studies including a prescribed Stepper walking program [39] and Nordic walking [45]. BMI did not change in participants of the study including a high-/moderate-intensity exercise program [42].

Body fat percentage (BFP) was measured in two studies, which included long-term regular walking [41] and Nordic walking [45]. Body fat percentage was significantly decreased in the intervention group in both studies. Furthermore, there was no significant change of BFP observed in the control group of these studies.

The waist-to-hip ratio (WHR) was assessed in two studies including a prescribed Stepper walking program [39] and Nordic walking [45]. WHR significantly improved in the intervention group in both studies; however, no difference in WHR was present in the control group.

#### 3.2.4. Muscle Strength

Muscle strength was measured in two studies, and it was assessed as handgrip strength and handgrip/body weight ratio. Handgrip strength was measured in the study analysing long-term regular walking and did not show any significant improvement post intervention [41]. In the same way, the handgrip/body weight ratio did not change in the participants of the prescribed Stepper walking program study [39].

#### 3.2.5. Endurance

Aerobic endurance, inspiratory muscle endurance and calf endurance were measured as a predictor of overall endurance in two studies. Inspiratory muscle endurance (PEND) was examined through the 2 min incremental threshold loading test (ITL) in the study including an aerobic exercise training program combined with the incentive spirometry [38]. The results showed a significant improvement of PEND in the intervention group, while no significant difference was observed in the control group. Aerobic endurance and calf endurance were measured in the Nordic walking study [45]. Aerobic endurance did significantly improve in the experimental group, while it did not change in the control group. Calf endurance did not show any improvement in any of the groups.

#### 3.2.6. Physical Activity

Physical activity level (measured using pedometer) and fatigue intensity in performing ADL and IADL were obtained to evaluate the physical activity in participants. Physical activity levels significantly increased in participants performing regular long-term walking [41]. Fatigue intensity did not change in any of the groups in the study including the prescribed Stepper walking program [39].

#### 3.2.7. Other Outcome Measures

These measures included the six-minute walk test (6MWT), exacerbation of obstructive airway disease (EOAD), reaction time, flexibility, Sit and Reach test, Get Up and Go test, the functional movement screen (FMS), quality of life (SF-36), Mini mental state exam (MMSE), Barthel index, Instrumental Activities of Daily Living (IADL) and Geriatric Depression Scale (GDS).

The six-minute walk test (6MWT) was assessed in the study with a prescribed Stepper walking program [39]. The results showed a significant increase in distance walked during the 6MWT in the experimental group. No significant differences of this distance were observed in the control group.

Exacerbation of obstructive airway disease (EOAD) was determined in the study including the prescribed structured physical activity program [44]. The experimental group was linked to a greater incidence of EOAD or any respiratory hospitalisation compared to the control group.

There is no information about the change of reaction time and flexibility post intervention in participants of the study including a high-/moderate-intensity exercise program [42].

The flexibility of the lumbar spine was determined using the Sit and Reach test, and it was measured in a total of two studies. There was a significant improvement of the flexibility of lumbar spine in 1 of 2 intervention groups in the study including the Nordic walking intervention. No significant difference was observed in the control group [45]. In the study analysing the long-term regular walking program, there was significantly improved flexibility of the lumbar spine in both the intervention and control group [41].

The results of the Get Up and Go test were significantly better in the intervention group in the study of regular long-term walking. The results in the control group showed no significant change [41].

The functional movement screen (FMS) as a predictor of future musculoskeletal injuries was assessed in the study combining the aerobic training with rhythmic functional movement. Both FMS and quality of life (QoL, SF-36) were evaluated in this study. There was a significant difference of FMS and QoL between the experimental and control group post intervention. Nevertheless, there was no significant time-by-group interaction of FMS or QoL in any group. However, FMS and QoL showed a significant difference in time as a result of the two-way repeated-measures analysis [43].

The Mini mental state exam (MMSE), Geriatric Depression Scale (GDS-15), IADL scale and Barthel index were determined in the study including the prescribed Stepper walking program. There was no significant change in MMSE, GDS-15, the IADL scale or the Barthel index in time in any of the groups [39].

### 3.3. Methodological Quality Assessment

As a result of the methodological quality assessment of the seven included studies using the RoB2 tool, 1 study was evaluated as being of some concern and 6 studies as being of high concern (Figure 2). The risk of bias in individual parts of all studies was determined by two independent observers based on an algorithm for answering the questions described in the Cochrane risk of bias tool version 2 (RoB2) [36].

In terms of the randomization process (D1), none of the studies mentioned whether the allocation sequence had been concealed properly. The fact that studies did not report any baseline imbalance between intervention groups led to the evaluation of the randomization process in all seven studies as being of some concern.

Regarding the deviations from intended interventions (D2), all seven studies were evaluated as being of some concern. Most studies did not provide any information about the blinding process, as well as it could not be determined whether there were any deviations that arose from the trial context. None of the studies mentioned the intention-to-treat (ITT) or modified intention-to-treat (mITT) analysis. For this reason, even if the study met the criteria for being of low risk in the first part, the lack of appropriate analysis (ITT, mITT) led to an increased risk of bias. Therefore, all seven studies had to be rated as being of some concern.

Regarding missing outcome data (D3), 1 study was assessed as being of low risk, 1 study as being of some concern and 5 remaining studies as being of high risk. One study provided sufficient evidence that the results were not biased by the missing outcome data. One study did not provide sufficient evidence that the missingness of the data for sure did not depend on the true value. The remaining five studies did not provide enough information about missing outcome data; therefore, they had to be evaluated as being of high risk.

In terms of the measurement of the outcome (D4), 6 studies were evaluated as being of low risk and 1 study as being of some concern. In six studies, there was no reason to believe that the method of measuring the outcome was inappropriate or that the assessment of the data was influenced by knowledge of the intervention. In one study, there was a possible influence of the knowledge of the intervention because, after the primary randomization, two intervention groups were divided according to usual physical activity into physically active and physically inactive participants.

To select the reported results (D5), all seven studies were evaluated as being of some concern because there was only one possible way in which the outcome domain could be measured, while there was no pre-specified analysis plan available to compare to in any of the studies.

Overall, 6 studies were evaluated as being of high risk and 1 study as being of some concern.

## 4. Discussion

The aim of this systematic review was to evaluate the effect of different types of walking on lung function (FEV_1_, FVC) in healthy seniors. Each of the analysed studies used spirometry as an evaluation tool to assess the lung function. None of the studies worked with seniors suffering from a cognitive deficit, which could bias lung function measurement. Walking as a way of improving the health of seniors has been studied already for a long time. The beneficial effect of walking on various body systems has been demonstrated in many studies over the years [25,46,47,48,49]. Many of the literary resources therefore agree that aerobic exercise in various forms, including walking, positively affects the lung function in seniors [37,40,50]. However, the results of previous studies on aerobic exercises and their impact on lung function have been inconsistent [50,51]. Furthermore, the analyse of studies in this review showed that the impact of walking on lung function differed across the studies. A more detailed analysis of the studies follows.

Fragoso et al. [44] conducted a multicentre randomized controlled trial to examine the effect of structured physical activity including walking compared to the control group, which received health education classes combined with short and gentle upper extremity stretching or flexibility exercises. There was no significant intervention effect on lung function (FEV_1_, MIP) and dyspnoea observed at 6, 18 and 30 months of follow-up. The authors hypothesized that the intervention focused on lower extremity function would improve the respiratory parameters in the elderly as pulmonary rehabilitation programs also include lower and upper limbs exercises, inspiratory muscle training and health education about respiratory diseases [52,53,54]. However, the mechanisms underlying dyspnoea and respiratory impairment probably were not addressed by the exercise design used in the study. Physical activity was also linked to a higher risk of exacerbation of obstructive airway disease (EOAD) or any respiratory hospitalization. The differences in lung function and dyspnoea did not accompany the higher rate of EOAD or any respiratory hospitalizations. The authors do not know the exact mechanism underlying the higher rate of respiratory hospitalizations, although they suggest that the higher rate of respiratory hospitalizations may be linked to frequent clinical monitoring of participants in the intervention group, which could lead to an early diagnosis of the respiratory dysfunction in participants. To conclude, structured physical activity and health education lessons did not improve lung function (FEV_1_, MIP) or dyspnoea severity in seniors over a planned intervention period of 24–42 months.

Huang et al. [42] aimed to assess the effect of a high-/moderate-intensity exercise program on lung function (FVC, FEV_1_) in healthy seniors. There was a significant improvement of both FVC and FEV_1_ observed in the high-intensity exercise group. The moderate intensity group experienced a significant change in FVC. No significant changes were observed in the control group. These findings suggest that the effect of a high-/moderate-intensity exercise program appears to be dose-related since both spirometry outcomes (FVC, FEV_1_) improved only in the high-intensity exercise group. Based on these findings, the authors suggest that the dosing of the physical intervention in regard to improving the lung function in the elderly should be at least 40 min, 3× a week, for a total of 10 weeks at the intensity level of 85–90% of maximal heart rate (HRmax). The moderate-intensity exercise group improved only in the FVC parameter, which is consistent with the study by Shin et al. [55]. Shin et al. [55] showed significant improvement in FVC, but no FEV_1_ in Korean seniors after 8 weeks of walking (50–60 min for 3x a week) at the intensity level of 40–60% of HRmax. The findings of Huang et al. [42] may have important clinical implications regarding the choice of suitable dosing of physical activity in the elderly in order to improve their lung function.

Shim et al. [43] investigated the effect of aerobic training with rhythmic functional movement on lung function in the elderly. Both the experimental and control group showed improvement in FVC, FEV_1_ and MVV. The experimental group performed aerobic exercise by gently connecting the functional movement with music in the form of dance, whereas the control group performed the functional movement to the beat without music. FVC, FEV_1_ and MVV showed a significant difference in time as a result of two-way repeated measures analysis. There was a significant time-by-group interaction in FEV_1_ and MVV. There was not a significant time-by-group interaction observed in FVC. The authors did not explain the specific mechanism underlying the time-by-group change of FVC, FEV_1_ and MVV. When comparing the experimental and the control group, the improvement of lung function in the experimental group was more positive than in the control group. It can be summarized that aerobic training combined with rhythmic functional movement both performed with or without music is an effective way to improve lung function in the elderly when performed 2x a week, 50 min a session, for a total duration of 8 weeks.

Kuo et al. [39] examined the effect of a prescribed Stepper walking program on lung function in older adults. The intervention group performed a regular moderate-intensity Stepper walking exercise (30 min, 2x a week, 8 weeks), whilst the control group kept and recorded their routine everyday activity. There was no significant improvement of FVC, FEV_1_ and FEV_1_/FVC observed in either of the two groups. Regarding physiologic health, the results showed the improvement of some parameters in the intervention group compared to the control group. These parameters included BMI, waist/hip ratio, 6MWT distance, time needed for 3 m Up and Go test and body fat percentage. The authors hypothesized that in order to improve lung function (FVC, FEV_1_), the experimental group would require a more intensive dosing of exercise. According to Shin et al. [55], the effective dosing of aerobic exercise should be at least 50–60 min a session, 3x a week, for 8 weeks at an intensity level of 40–60% the target heart rate.

El-Kader et al. [38] conducted a study including aerobic exercise training combined with incentive spirometry. The mean values of VC, FEV_1_ and other respiratory parameters were significantly increased post intervention in the experimental group, but not in the control group. These findings are consistent with previous studies including incentive spirometry training in the elderly [56,57]. The increase of VC and FEV_1_ might be related to the enhanced strength of the respiratory muscles. More effective use of the diaphragm during expiratory movement and better coordination of the respiratory muscles in general probably led to the increase in FEV_1_. Respiratory exercise using incentive spirometry also enhances the production of pulmonary surfactant, increases lung compliance and reduces the airway resistance [58]. In conclusion, treadmill aerobic exercise (30 min, 3x a week, 3 months, 60–75% HRmax.) combined with incentive spirometry (5 min, 5x a week, 3 months) is an effective way to achieve better lung function in the elderly.

Šokeliene et al. [45] conducted a study analysing the effect of Nordic walking on lung function and physical fitness in the elderly. Participants were divided into 2 intervention groups and 1 control group. The two experimental groups (physically active group—PA, physically inactive group (PINA)) differed in the amount of normal physical activity of the participants, which was initially assessed using questionnaires. Results showed a significant improvement of VC post intervention in both experimental groups, but no improvement in the control group. The improvement of VC was more positive in the PINA group than in the PA group. However, the authors did not explain the mechanism underlying the improvement of VC in the 2 experimental groups, nor did they explain the difference in the change of VC between the 2 experimental groups. Various studies showed that Nordic walking is associated with increased oxygen and calorie consumption compared to regular walking [59]. Therefore, these findings may explain the improvement of VC in both intervention groups. Nevertheless, the reason for the significant improvement in VC in the PINA group was not elucidated in terms of this study. To conclude, Nordic walking is an effective exercise to improve lung function in the elderly when performed 50 min, 2x a week, for 12 weeks at a level of 50–85% HRmax.

Saygin et al. [41] aimed to investigate the effect of long-term regular walking on some physical functions in the elderly. The experimental group performed regular walking for 45–50 min, 3x a week, for 6 months. The control group continued their regular lifestyle activities. There was significant improvement of FVC post intervention in the experimental group. No changes were observed in the control group. These findings support the beneficial effect of walking on the individual’s overall health and quality of life, which was already proven in various studies [60,61]. Lee et al. [50] compared the effect of one hour of simple walking in the natural and urban environment in elderly Korean women. The results of the study showed a favourable effect of simple walking in the natural environment on selected spirometry parameters (FEV_1_, FEV_6_). However, a beneficial effect on lung function was not observed in the group walking in the urban environment. Therefore, we can reconsider the appropriate dosage of simple walking to achieve a beneficial effect on lung function in the elderly. There is also the issue of air pollution, as in a study by Lee et al. [50], a beneficial effect on lung function was observed also after 1 h, but only in a group walking in a natural environment.

Taking into account that the search timeframe was broad and that four of the main scientific databases were searched, our results highlight that the literature regarding spirometric respiratory parameters in relation to walking programs is still scarce. Considering that the number of older adults is expected to more than double by 2050 and triple by 2100 [62], our results can help clinicians have more information on how to implement walking in this growing population of older adults. Moreover, walking is a low-cost and simple exercise to implement, and spirometry is considered, among the different methods of lung volumes measurements, the most common pulmonary function test [63,64]. However, the number of retrieved studies was limited; therefore, this review can set the lines for future research, which could implement randomized controlled trials regarding older adults and walking programs in relation to respiratory parameters.

This systematic review has several limitations. Limitations apply both to the sample of analysed studies, but also to the individual studies separately. Limitations applying to the seven studies as a whole follow. The number of studies analysed in this review was relatively low. All studies included relatively healthy and fit seniors, except for one study, which included seniors with mobility limitations [44]. Therefore, it is not possible to apply the results of this systematic review uncritically to the entire senior population.

Three of the seven studies formulated their own limitations including a small sample size [39,42], participants with mobility limitations [44], exclusively female participants or their predominance in the sample [39,43], recruitment of participants from a single facility [39] or the possibility of bias in the results regarding the control group, which received some form of intervention [42,43,44]. It is therefore appropriate to consider and understand the limitations of the studies before interpreting these results and applying them to clinical practice.

## 5. Conclusions

The present review analysed studies that focused on the effect of different types of walking on respiratory outcomes in healthy seniors. Our results showed that choosing a suitable dosing of minutes and days and a target heart rate for performing physical activity may have important clinical implications regarding the lung function of older adults. Thus, these results should be taken with caution due to the limitations, and further reviews with larger sample sizes would be recommended.

## Figures and Tables

**Figure 1 ijerph-19-07995-f001:**
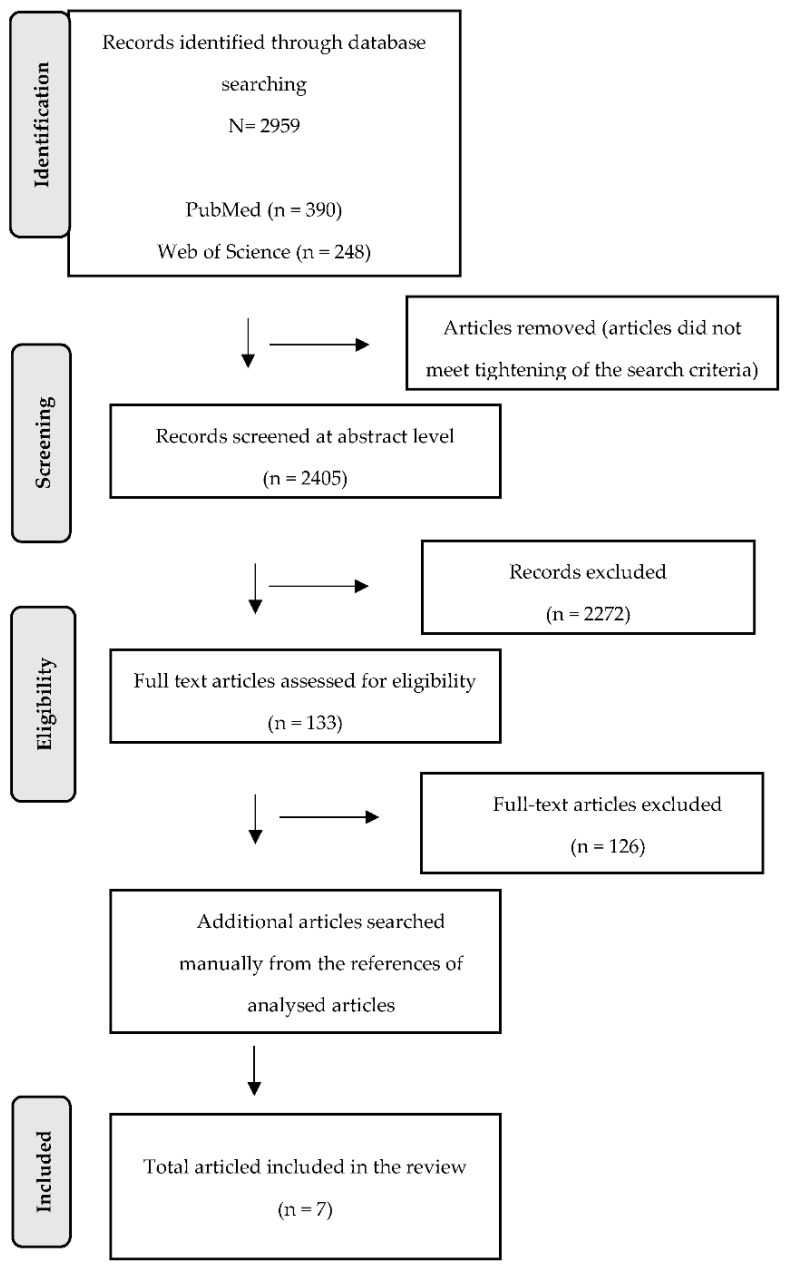
Flow diagram illustrating the selection process.

**Figure 2 ijerph-19-07995-f002:**
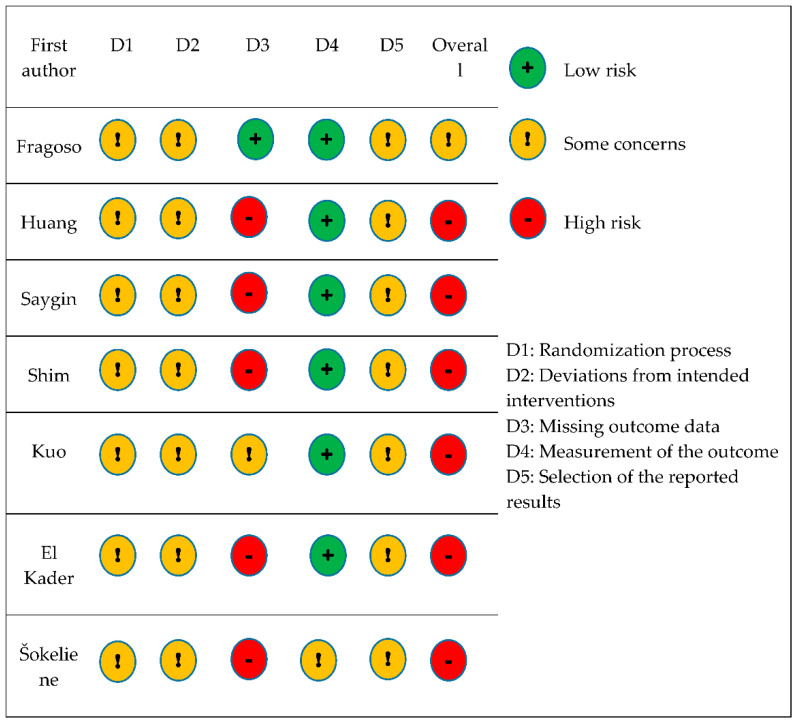
Risk of bias summary of the included studies.

**Table 1 ijerph-19-07995-t001:** The PRISMA statement.

Database	Keywords	Total Number of Results	Number of Articles in English Published between 2002–2022
Web of Science	ALL = (elderly) OR ALL = (seniors) OR ALL = (aged) OR ALL = (“older adults”) AND ALL = (walking) OR ALL = (“aerobic training”) AND AB = (spirometry) OR AB = (FVC) OR AB = (FEV_1_) OR AB = (“lung function”) AND ALL = (“randomized controlled trial”) OR ALL = (RCT) OR ALL = (randomized)	248	202
PubMed	(elderly) OR (seniors) OR (aged) OR (“older adults”) AND (walking) OR (“aerobic training”) AND (spirometry [Title/Abstract]) OR (FVC[Title/Abstract]) OR (FEV_1_[Title/Abstract]) OR (“lung function”[Title/Abstract]) AND (“randomized controlled trial”) OR (RCT) OR (randomized)	390	290
Scopus	ALL (elderly) OR ALL (seniors) OR ALL (aged) OR ALL (“older adults”) AND ALL (walking) OR ALL (“aerobic training”) AND ABS (spirometry) OR ABS (fvc) OR ABD (fev1) OR ABS (“lung function”) AND (ALL (“randomized controlled trial”) OR ALL (rct) OR ALL (randomized)	1652	1279
EBSCO Essentials	“AND elderly OR seniors OR aged OR “older adults” AllFields AND walking OR “aerobic training” AllFields AND spirometry OR FVC OR FEV1 OR “lung function” Abstract AND “randomized controlled trial” OR RCT OR randomized AllFields“	669	634

Abbreviations: FVC: forced vital capacity; FEV_1_: forced expiratory volume in 1 s; RCT: randomized controlled trial.

**Table 2 ijerph-19-07995-t002:** Eligibility criteria for study selection.

PICOS	Inclusion Criteria	Exclusion Criteria
Population	Studies including seniors aged 60 and above	Studies not including seniors aged 60 and above
Intervention	Different types of walking	Studies not including different types of walking as intervention
Comparator	Studies including a comparison group (control group, placebo group)	NA
Measured outcome	Spirometry	Studies not including spirometry
Study design	RCT	Cross-over studies, reviews, non-randomized controlled trials, preliminary studies, studies with no control group

Abbreviations: NA: not applicable; RCT: randomized controlled trial.

**Table 3 ijerph-19-07995-t003:** Keywords.

PICOS	Keywords
P	(“elderly” OR “seniors” OR “aged” OR “older adults”)
I	(“walking” OR “aerobic training”)
C	NA
O	(“spirometry” OR “FVC“ OR “FEV_1_” OR “lung function”)
S	(randomized controlled “trial” OR “RCT” OR “randomized”

Abbreviations: NA: not applicable; FVC: forced vital capacity; FEV_1_: forced expiratory volume in 1 s; RCT: randomized controlled trial.

**Table 4 ijerph-19-07995-t004:** Main characteristics of included studies.

First Author	Fragoso	Huang	Saygin	Šokeliene	Shim	Kuo	El Kader
**Year**	2016	2005	2015	2011	2019	2018	2013
**N**	1635	45	30	41	30	36	40
**Female**	1098	not stated	30	30	not stated	30	20
**Male**	537	not stated	0	11	not stated	6	20
**Mean Age** **(±)**	78.7 (5.2) I.G. 79.1 (5.2) C.G.	85.3 (2.5) 82. (3.1) I.G. 82.9 (3.0) C.G.	74.28 (8.78) I.G. 75.53 (7.21) C.G.	65 (5) I.G. 65(5) C.G.	75.33 (5.31) I.G. 74.40 (3.24) C.G.	68.93 (3.81) I.G. 70.38 (5.22) C.G.	67.27 (5.05) I.G. 69,18 (4.13) C.G.
**Intervention**	Structured physical activity	High/ moderate intensity exercise program	Long-term regular walking	Nordic walking	Aerobic training with rhythmic functional movement	Prescribed Stepper walking program	Aerobic Exercise Training + Incentive Spirometry
**Duration**	24–42 months	10 weeks	6 months	12 weeks	8 weeks	8 weeks	3 months
**Comparison Group**	Control group	Control group	Control group	Control group	Control group	Control group	Control group
**Outcome** **Measurement**	FEV_1_, maximal inspiratory pressure (MIP), exacerbation of obstructive airways disease (EOAD)	FVC, FEV_1_, diastolic blood pressure, systolic blood pressure, resting heart rate, flexibility, reaction time, height, weight	FVC, Sit and Reach test, handgrip strength, Get Up and Go test, physical activity level, body fat percentage, height, weight	VC, aerobic endurance, calf endurance, Sit and Reach test, weight, height, waist/hip ratio	FVC, FEV_1_, maximal voluntary ventilation (MVV), The functional movement screen (FMS), quality of life (SF-36)	FVC, FEV_1_, 6MWT, fatigue intensity, extremity muscle power, BMI, MMSE, Barthel index, IADL scale, GDS-15, waist/hip ratio, body fat %, handgrip/body weight ratio	VC, FEV_1_, MVV, PImax., PEmax., inspiratory muscle endurance
**Study** **Design**	RCT	RCT	RCT	RCT	RCT	RCT	RCT

## Data Availability

Not applicable.

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
