# Peer review of "Influence of Walking as Physiological Training to Improve Respiratory Parameters in the Elderly Population"

_ijerph, 2022, doi:10.3390/ijerph19137995_

Round 1

Reviewer 1 Report

The paper is interesting and I think the Material and Methods section has been described very clearly. Congratulations.

However, the Conclusions section raises some doubts in my opinion, specially due to the limitations shown above. Personally I think there are too many ideas to take into account. They are not appreciated in this version. Therefore, I think it is certainly brief. Please, consider expanding this section. For example, would it make sense to make any reference to possible differences in the age groups studied?

Author Response

The paper is interesting and I think the Material and Methods section has been described very clearly. Congratulations.

Dear reviewer, thank you for the time you have spent on our manuscript and thank you very much for your evaluation.

However, the Conclusions section raises some doubts in my opinion, specially due to the limitations shown above. Personally I think there are too many ideas to take into account. They are not appreciated in this version. Therefore, I think it is certainly brief. Please, consider expanding this section. For example, would it make sense to make any reference to possible differences in the age groups studied?

Thank you for your opinion and recommendations. According to our previous experience with articles in MDPI journals, the conclusion was deliberately written briefly. However, we have added a section of the discussion, marked in red, where we draw attention to your recommended aspects.

Reviewer 2 Report

Although the methodology is correct, the number of articles is very limited. This is due to the very restrictive inclusion criteria.

There is ample evidence of the influence of walking in relation to various physiological parameters of the elderly population. Therefore, I suggest the authors carry out a review on this type of physical activity (walking) that allows for a systematic review and, above all, a relevant meta-analysis. Where effect size, standardized mean difference, or correlation can be analyzed.

Although the method applied in the systematic review that the authors propose is correct and fine. With seven papers, a review of relevance cannot be carried out and therefore publishing the data in a journal of this prestige does not seem appropriate to me. For all these reasons, I am going to propose that the paper be rejected.

Author Response

Although the methodology is correct, the number of articles is very limited. This is due to the very restrictive inclusion criteria.

Thank you for taking the time to read our manuscript. Thank you also for your opinion on the methodology. We do not agree with your opinion, because our procedure was correct, the criteria were set intentionally strictly precisely so that the methodological procedure was correct.

We  worked in our study according to the Preferred Reporting Items for Systematic Reviews and Meta-Analysis checklist (PRISMA), we followed the guidelines of the Cochrane Handbook for Systematic Reviews of Interventions version 6.3. The eligibility criteria were based on the PICOS framework, and the quality of each study included in the review was assessed using the Cochrane risk of bias tool version 2.  All these procedures ensure quality of the process and therefore, that the results of our study, although limited by number of studies, are presented with scientific rigour.

If we look at published articles in IJERPH, recently, we find a large number of reviews that have worked with a lower number of included studies than in the case of our study - for example: https://doi.org/10.3390/ijerph19116764 - 5 papers; https://doi.org/10.3390/ijerph19127202 - 6 papers; https://doi.org/10.3390/ijerph19127061 - 7 papers; etc.

There is ample evidence of the influence of walking in relation to various physiological parameters of the elderly population. Therefore, I suggest the authors carry out a review on this type of physical activity (walking) that allows for a systematic review and, above all, a relevant meta-analysis. Where effect size, standardized mean difference, or correlation can be analyzed.

Although the method applied in the systematic review that the authors propose is correct and fine. With seven papers, a review of relevance cannot be carried out and therefore publishing the data in a journal of this prestige does not seem appropriate to me. For all these reasons, I am going to propose that the paper be rejected.

Of course, we know that there are a number of studies evaluating the effect of walking on various parameters, our intention was to evaluate the effect on FEV1 parameters. There are not many such studies yet and this is the reason for the small number of studies that met our criteria. It is our results that point to the need for further research, which we consider to be a clear result. In the conclusions, we present only the recommendations that follow from our studies. In scientific work, it is not possible to change the criteria "on the fly" in order to achieve the results we expect; we consider our approach to be correct in the study presented and we will take your proposal to provide further studies. Additions have been made to the discussion chapter.

We are very sorry for your conclusion, because we believe that our study brings a lot of new things, despite the fact that it only worked with seven studies. We do not consider the reason for the exclusion of our manuscript due to 7 studies to be fair, because if you look in the journal, you will only find reviews in the last month that have worked with an even lower number of studies and have been accepted (see above). Not only for this fact, we would be very happy if you would evaluate our study in the light of the goals we set and not the goals that you would recommend to formulate.

Reviewer 3 Report

The reviewing process was shown and explained it clearly. 

The section of keywords in the Table 1 should be rewritten. So many parentheses ((( ))). Very confusing.

Table 4 can be summarized in 1-2 pages. 

The figure (I assume Figure 1) needs the title and explanation. Further, the marks of indicating the risk level were not easy to understand and the marks should be simple (, ×). D1 to D5 should be put in the figure.

Only 7 studies were met the criteria, but it may not be enough to conduct a systematic review. Overall, the authors explained each research methods well. However, this is a systematic review study so that the authors should focus on what they reviewed (found).

Line 438, 491, 504 & 550: Erase the lines.

Author Response

The reviewing process was shown and explained it clearly

Dear reviewer, thank you for your time in reading our manuscript and thank you for your assessment.

The section of keywords in the Table 1 should be rewritten. So many parentheses ((( ))). Very confusing.

Thank you for the recommendation, we unequivocally agree with your opinion, the brackets have been removed. The text now looks clearer.

Table 4 can be summarized in 1-2 pages.

Thank you for the recommendation, we have modified the table so that it is now on 2 pages.

The figure (I assume Figure 1) needs the title and explanation. Further, the marks of indicating the risk level were not easy to understand and the marks should be simple (◯, △, ×). D1 to D5 should be put in the figure.

Thank you for your recommendation, we have reworked the figure and explained the symbols.

Only 7 studies were met the criteria, but it may not be enough to conduct a systematic review. Overall, the authors explained each research methods well. However, this is a systematic review study so that the authors should focus on what they reviewed (found).

We concur with you, that the number of retrieved studies is not high, however, there are previous systematic reviews that have been published with similar number of reviewed articles (for example -  published articles in IJERPH last month: which are also https://doi.org/10.3390/ijerph19116764 - 5 papers; https://doi.org/10.3390/ijerph19127202 - 6 papers; https://doi.org/10.3390/ijerph19127061 - 7 papers; etc.)

Moreover, our study was written according to the Preferred Reporting Items for Systematic Reviews and Meta-Analysis checklist (PRISMA) and followed the guidelines of the Cochrane Handbook for Systematic Reviews of Interventions version 6.3. In addition, the eligibility criteria were based on the PICOS framework, and the quality of each study included in the review was assessed using the Cochrane risk of bias tool version 2.  All these procedures ensure quality of the process and therefore, that the results of our study, although limited by number of studies, are presented with scientific rigour.

Nevertheless, we have highlighted the number of retrieved articles as a limitation, in order to take this into account when interpreting our results.  We quote: “This systematic review has several limitations, being the main one the small sample size, thus the retrieved studies were only 7. Limitations apply both to the sample of analyzed studies, but also to the individual studies separately. Limitations applying to the 7 studies as a whole follow. The number of studies analyzed in this review is relatively low. All studies included relatively healthy and fit seniors, except for 1 study which included seniors with mobility limitations. Therefore, it is not possible to apply the results of this systematic review uncritically to the entire senior population.”

In relation to our results, following your suggestion we have added a paragraph to focus on what we reviewed and giving it more strength, though considering the limitations (in discussion part)

Line 438, 491, 504 & 550: Erase the lines.

Thank you for the recommendation, the lines have been removed to make up the text.

Round 2

Reviewer 2 Report

The paper has improved, but it still has the limitations that I mentioned in the previous review.

Reviewer 3 Report

The authors have thoroughly considered the suggestions and criticisms from the reviewers and revised the manuscript accordingly.